# Application of video surveillance in preclinical safety studies in canines: Understanding the interobserver reliability and validity to recognize clinical behavior

Eline Eberhardt[1], Fetene Tekle[2], Greet Teuns[3], Jill Witters[4], Bianca Feyen[1], Sarah De Landtsheer[4], Ivan Kopljar (ORCID)[1]*

1 Nonclinical Safety & Submissions, Preclinical Sciences and Translational Safety, Johnson & Johnson Innovative Medicine R&D, Beerse, Belgium, 2 Statistics & Decision Sciences, JRD Global Development, Johnson & Johnson Innovative Medicine R&D, Beerse, Belgium, 3 Translational Pharmacokinetics Pharmacodynamics & Investigational Toxicology, Preclinical Sciences and Translational Safety, Johnson & Johnson Innovative Medicine R&D, Beerse, Belgium, 4 Scientific & In vivo Strategies, Preclinical Sciences and Translational Safety, Johnson & Johnson Innovative Medicine R&D, Beerse, Belgium

* IKopljar@its.jnj.com

## Abstract

Preclinical *in vivo* studies are critical to identify potential adverse effects of drugs under development. However, a significant number of drug candidates are terminated during human clinical trials due to unexpected adverse events which were not predicted or detected in preclinical studies. Video surveillance can be a valuable tool to reduce the risk of missing and/or misclassifying adverse clinical observations (COs) in animals. To explore the applicability of detecting COs on video, the agreement between observers (reliability) and agreement between observers and experts (construct validity) of 13 important COs was evaluated. The reliability was investigated by evaluating the interobserver agreement between 23 observers with different experience levels and primary roles on defined COs and normal behavior, recorded on video during preclinical studies in canines. The validity was investigated by comparing the observers' assessments to the ground truth confirmed by three experts. This investigation showed a substantial reliability and validity of the observers' assessments without significant differences between experience levels or primary roles. Normal behavior was challenging to recognize (56% correct), while half of the COs appeared straightforward to identify with a validity of ≥ 90%: salivation, aggressiveness, circling, vomiting, head shaking and convulsions. Other COs were more challenging to detect with lowest scores for limb stiff/hypertonia, tremors and excitation. Regardless of experience-level, observers missed very few COs. This investigation showed the complexity when multiple COs occurred simultaneously, as well as the limitations of differentiating between visually similar COs (tremors vs. twitches; limping vs. limb stiff) on video without the possibility of *in-person* observation. Given the substantial overall reliability and validity, it is concluded that clinical canine behavior

**Data availability statement:** All relevant data are within the manuscript and its Supporting Information files.

**Funding:** The work presented in this paper was carried out in the framework of project grant HBC.2021.1126 to IK, funded by the government Flanders Innovation & Entrepreneurship (VLAIO) agency; http://www.vlaio.be. The funders had no role in study design, data collection and analysis, decision to publish, or preparation of the manuscript.

**Competing interests:** All authors have read the journal's policy and the authors of this manuscript have the following competing interests: all the authors were employed by company Janssen Research and Development, Janssen Pharmaceutical Companies of Johnson & Johnson. This does not alter our adherence to PLOS ONE policies on sharing data and materials.

can be accurately detected on video by trained observers. This permits more objective and quantifiable monitoring of animal behavior and application of computer vision for future automatic monitoring of canine studies.

## 1. Introduction

A critical step in nonclinical drug development is to determine and to de-risk the potentially adverse effects of a novel drug candidate, by combining *in silico, in vitro* assays and *in vivo* rodent and non-rodent animal models – amongst which the Beagle dog is widely used and the focus of this paper. Adverse effects are typically evaluated in toxicology studies in which the target organs are identified as well as the dose dependence of the adverse effect(s), their relationship to exposure and potential reversibility [1,2].

During preclinical *in vivo* safety studies (e.g., toxicology and safety pharmacology), all tested species are observed for behavioral changes and clinical observations (COs) to detect potentially adverse effects. The observations are performed by trained lab technicians and scientists who check the animal's behavior in person in the animal unit. However, these evaluations are limited in time (only specific timeframes during working hours), subjective and introduce human interference – implying that adverse events can be missed, that the observations are potentially altered by human presence and that there is not always a quantitative analysis feasible. Indeed, a significant proportion of drug candidates are terminated in clinical study phases due to unexpected adverse clinical events in human subjects [3–5]. This might be partially attributed to COs that were missed (not detected) during preclinical study of animals. To reduce this risk, video surveillance was explored as a method to enable more objective and quantifiable monitoring of animal behavior without the need for human interference. Moreover, video monitoring enables continuous observation of animal health and wellbeing and therefore increases the refinement of animal studies as part of the fundamental 3R principles (Replacement, Reduction and Refinement) [6].

When describing animal behavior, it is well known that observers may give different interpretations of the same behavior related to several causes, *e.g.,* function and experience of the observer, different interpretation of the behaviors' definition, different detection or relevance given to available cues, observers trusting on their 'feeling' instead on visible signs, observers' subconscious bias towards what they expect to see and mistakes in what the observer thinks he/she saw [7–9]. This makes the interobserver agreement (IOA) or reliability a crucial aspect in all behavioral research. IOA measures the extent to which different observers agree on the occurrence of a behavior when evaluating the same subject during the same observation period [7,8]. Funder proposed four categories to differentiate causes of disagreement between observers: *i) good judge:* some people might be better observers than others; *ii) good target:* some subjects are possibly easier to observe than others; *iii) good trait:* some behaviors are easier to observe than others; and *iv) good information*: certain kinds of information (or definitions) make observations more accurate [7,10]. Unfortunately,

in observational studies of animal behavior, not all four categories of IOA are always addressed [8,11] with focus mainly on the *good judge* which can be affected, amongst others, by practice, experience and training [7,8].

IOA on canine behavior has been investigated previously, in particular on behavior and welfare of shelter dogs in view of the likelihood of adoption [9,12–14]. IOA was also included in a number of publications that investigated the performance of working dogs [7,15–17], pet dog behavior [18–20], and the detection of veterinary-related abnormalities [21–25]. A number of publications go more into detail on Funders' categories of disagreement. For the *good judge* category, it is expected that the IOA increases with the level of the observers' experience. This is shown in several studies [7,9,21,22], although other publications point out that novice observers can also obtain satisfactory agreement with experts [7,18,26]. A variety of research in canines touches base on the *good trait*, to get an insight in which behaviors are harder to identify than others [7,9,12,13,15,17,18,20–22]. Clark *et al.* studied the effect of *good information* and investigated whether observer ratings were improved upon adapting the rating scale with verbal cues [16]. With regard to the use of video footage, Lazarowski *et al.* demonstrated a high agreement between live scorers and observers who scored on video footage (mobile camera) of the same observational timepoint [15].

In the present work, the focus was on the reliability (agreement between observers) and construct validity (agreement between observers and experts) of the observers' assessments of videos from fixed-positioned cameras. The reliability was assessed by examining the level of agreement among 23 observers with varying experience and primary roles (technicians, veterinarians and study scientists) on 13 pre-defined COs and normal behavior, originating from different preclinical safety studies. The construct validity of the observers' assessments was investigated by comparing these to the observations made by three experts on the same videos. Construct validity is referred to as simply 'validity' throughout the manuscript. Three categories of Funders were evaluated: the impact of the observers' experience and primary role (*good judge*), the influence of the behavior itself (*good trait*) and whether some videos were more difficult to judge than others (*good target*). To our knowledge, this is the first research with extensive investigation on the interobserver reliability and validity on canine behavior within the preclinical safety field and the results may have an important impact on describing and interpreting certain COs from video data.

## 2. Materials & methods

### 2.1. Clinical observations (COs)

Thirteen COs were selected for the IOA as they *i)* are regarded as important observations during preclinical *in vivo* studies; *ii)* can be identified on short video snippets; and *iii)* do not require additional physical examination for their confirmation. Selected COs with their definition are shown in Table 1.

### 2.2. Selection of videos

Video surveillance cameras (AXIS P3235-LVE or AXIS P3245-LVE) were positioned in front view and outside of the animal housing space at approximately 1.2m height. Video data was captured at 25 frames per second. For the current research, a single expert initially selected 73 videos from which 70 videos were retained for the IOA analysis following the reviewing process by all three experts. These 70 video snippets had a duration ranging from 8 to 67 seconds each (excluding one outlier of 6 min 40sec); and represented a total of 13 various COs as observed and reported by the lab technicians during cage side observations. All the data used in this work were solely sourced from historical video footage of animal studies (toxicity and mechanistic studies) previously performed in house and originate from two different experimental rooms with the same housing and camera setup. The studies were conducted in accordance with the European Directive of 2010 (2010/63/EU) on the protection of animals used for scientific purposes and the Belgian and Flemish Region implementing legislation and were conducted in and have been approved by the ethics committee on Animal Experiments of the Research Center of Janssen Research & Development, a division of Janssen Pharmaceutica NV, located in Beerse, Belgium which is accredited by AAALAC (https://www.aaalac.org/) since 2004. All animals were group

**Table 1. List of 13 clinical observations of interest.**

| Clinical Observation | Definition |
|---|---|
| Aggressiveness | Any threat or harmful behavior directed towards another individual or group; commonly includes body language or threat displays such as hard stare, growling, barking, snarling, lunging, snapping, and/or biting. |
| Anxiety | Psychological and behavioral state induced by a threat to wellbeing or survival, either actual or potential. Characterized by increased arousal, expectancy, autonomic and neuroendocrine activation, and specific behavior patterns such as tucked tail, lip licking, barking or howling, shivering, inability to settle, running away and/or cowering in a corner. |
| Ataxia | Inability to coordinate voluntary muscle movements, loss of balance. |
| Circling | Continuous movement in a circular direction. |
| Convulsions | *Clonic type*<br>Convulsion with alternate contraction and relaxation of voluntary muscles.<br>*Tonic type*<br>Persistent contraction and spasm of a set of voluntary muscles. Sudden stiffening of the fore or fore and hind legs which are stretched in backward direction with greatly increased muscle tone.<br>*Miscellaneous type*<br>All other phenotypes of convulsions, *e.g.,* praying: a sitting-up seizure in which the forelimbs are held together or crossed in a posture resembling prayer. |
| Excitation | The state of being emotionally aroused. Typical behavior patterns include an open mouth with tongue hanging out, inability to settle, panting, jumping up and down, vocalizing incessantly, teeth chattering, and/or full body shaking. |
| Head shaking | Moving the head up and down or from left to right. |
| Limb stiff/ Hypertonia | Increased muscle tone, and lack of flexibility. |
| Limping | A lame walk with a yielding step, asymmetric gait. |
| Salivation | Excretion of saliva from the salivary glands. |
| Tremors | Involuntary, purposeless, oscillatory movements which result from the alternate contraction of muscle groups. |
| Twitches | Brief, coarse, involuntary muscle contractions which cause the animal to abruptly jerk or twitch its limbs and/or body. |
| Vomiting | Forceful, voluntary or involuntary ejection of the stomach contents. |

For each CO, the corresponding definition is stated as defined within our preclinical safety department.

housed. Individual housing during studies occurred only during feeding for a duration of ≤4hr. They were under the care of trained individuals with veterinary oversight and received appropriate veterinary care, if needed, for clinical symptoms that they might have developed.

## 2.3. Defining the ground truth and optional observations

To evaluate the construct validity and reliability of observer's assessments, first the ground truth was determined together with the criterion validity. The latter was required to validate that experts were able to identify clinical observations on video compared to the *in-person* observations which were considered as the 'gold standard' [27].

A total of three experts determined the ground truth. They have a combined 48 years of experience in assessing behavior and COs in canines within preclinical safety studies, have different primary functions and work in separate departmental groups (*in vivo* sciences, safety pharmacology and toxicology). Expert 3 selected the video snippets starting from a master list of clinical observations registered on specific timepoints during *in-life* study evaluations by lab technicians (experience ranging from 5–40 years). Expert 3 checked randomly videos matching the timestamp for the specific CO. In case expert 3 confirmed the CO, the video snippet was retained for further blinded and independent evaluation by experts 1 and 2 to define the primary and optional observations. In general, snippets originated from different studies and/ or animals.

When all three experts agreed on the observed CO(s) within a video, the CO(s) was marked as a primary observation, which composed the ground truth. The ground truth thus consisted of observations that were previously noted during

 

*in-person* monitoring and confirmed independently by all three experts on video recordings. In two videos, all three experts agreed on an additional CO that was not registered during *in-person* monitoring (limping, excitation). In this case, both COs were kept as primary observation. If only two out of three experts described a certain CO on video and a consensus could not be reached between the three experts after face-to-face discussions, those COs were marked as optional. In case a certain CO was only noted by one expert, this CO was rejected. Of the initial 73 selected videos by expert 3, a total of three videos were rejected as both experts 1 and 2 did not confirm the COs (assumingly) present in those videos. The final selection of 70 videos had at least one primary observation, whereas 15 videos had at least one additional optional observation. Up to three primary and two optional observations were described per video.

Table 2 shows the distribution of primary and optional COs on the selected 70 videos. A total of 13 different primary (including normal behavior) and nine optional observations were identified; twitches were present as an optional observation only as, for all videos, only two out of three experts found agreement. Eleven videos showed normal behavior (no clinical observation) as a primary observation. On the 59 remaining videos, a total of 70 primary and 17 optional COs were defined following evaluation by the experts. Irrespective of the presence of optional CO(s), 49 videos showed only one primary CO, while 10 videos showed two or more primary COs. The majority of the videos included sound (54 out of 70). For five COs (anxiety, circling, convulsions, excitation, and head shaking), all videos included sound. The effect of sound as an aid to correctly identify COs was not investigated in this work.

After agreement on the ground truth, both blinded experts 1 and 2 were asked to give a score on how easy or difficult it was to detect the primary or optional observation(s) in the 70 videos: *(0)* not detectable; *(1)* difficult to detect; *(2)* easy to detect.

## 2.4. Observers and IOA Questionnaires

A total of 23 observers with variating levels of experience and different primary roles participated in this research. All of them were working within the department of Preclinical Sciences and Translational Safety and have been involved in at least one preclinical safety study with canines. The recruitment period started on 24 October 2022 and ended with the

**Table 2. Overview of the 13 different COs and normal behavior (no CO).**

| Behavior | Primary Observation | Optional Observation | N° videos |
|---|---|---|---|
| *No Clinical Observation* | 11 | 0 | 11 |
| Aggressiveness | 4 | 2 | 59<br>− 49: 1 primary CO<br>− 10: >1 primary CO |
| Anxiety | 5 | 0 | |
| Ataxia | 10 | 3 | |
| Circling | 5 | 0 | |
| Convulsions | 6 | 1 | |
| Excitation | 6 | 1 | |
| Head shaking | 5 | 0 | |
| Limping | 5 | 0 | |
| Limp Stiff/ Hypertonia | 2 | 3 | |
| Salivation | 6 | 1 | |
| Tremors | 10 | 2 | |
| Twitches | 0 | 3 | |
| Vomiting | 6 | 1 | |
| Total | 81 | 17 | 70 |

Overview of the different COs with the number of videos in which the behavior was seen as 'Primary Observation' and 'Optional Observation'. Note that twitches were only present as an optional observation. The third column shows the total number of videos containing either normal behavior or a CO/ multiple COs.

distribution of the questionnaires on 18 November 2022. A written informed consent was obtained from all participating observers in which they agreed to their answers being included in the analyses.

The primary role was assigned to the observers based on their background and functional responsibilities. Observers were asked to evaluate their level of experience themselves, ranging from minimal to high. The experience levels were defined as following: *minimal:* observer was familiar with some of the listed COs and/or had limited experience in observing these in canines; *moderate:* observer was familiar with most/all of the COs, and was in general able to identify all of these and had observed most of them in canines; *high:* observer was familiar with all the listed COs, was able to identify all of these and had observed all of these in canines.

Prior to sending out the IOA questionnaires, observers received a refresher training by the experts on the definition of all 13 COs, illustrated with video examples (that were not included in the IOA dataset). Instructions on how to complete the IOA questionnaire were also given at the end of the training.

The IOA videos were randomized and numbered from 01–70. Observers watched the videos individually at their own pace. They were allowed to rewind, pause and re-watch the videos, but were instructed not to discuss the videos with other participants. The IOA questionnaire consisted of a Microsoft Excel file with one row per video and four columns. Per video, participants had to select at least one and up to four different observations from a pre-defined drop-down list, which included the 13 COs, 'no clinical observation' and 'other clinical observation, not listed'. The latter was added for completeness, but was not included in the analysis as none of the videos contained any COs outside of the pre-defined set. Assessment was always done on a single animal. In case there was more than one animal present in a video, an identifier was stated in the Excel file (*e.g.,* animal left, yellow collar,..). Per video, a comments column was also included in the questionnaire (free-text field) to allow observers to elaborate on their findings and write down potential doubts. Participants were given three weeks to watch the video snippets and complete the questionnaires. Each observer was assigned a unique letter code (A-W) that was further used in the analysis.

## 2.5. Quality control & incorporation of relevant comments

Upon receiving the completed IOA questionnaires from all 23 observers, a quality check was performed. The following items were checked on the data: *i)* empty response fields, *ii)* obvious mix-ups in case multiple animals were present in the video, and *iii)* conflicting combination of both 'No CO' and a CO entered for the same video. In case an observer's answer did not pass the quality check, the observer was contacted and asked to re-check that specific video(s) and to confirm or re-evaluate the response. The final response was then corrected in the completed IOA questionnaire, using a track change function.

A second part in the processing of the completed questionnaires was the incorporation of the comments section. Some observers entered a correct answer in the comments field, while not choosing it via the drop-down menu among the list of the COs. An example of a relevant comment for a video showing CO x was: '*It seems that the animal is showing CO x, but I would need a longer video fragment to be sure*'. An example of a relevant comment of an observer who entered a CO for a video containing solely normal behavior was: '*Not sure, this could also be normal behavior*'. From all comments, the relevant comments were compiled into a separate 'corrected answer' Excel file in which the original answers were replaced by or completed with the correct answer(s) described in the comments.

## 2.6. Statistical analysis

In this research, reliability is referred to when investigating the agreement between the observers, and (construct) validity when investigating the agreement between observers and the ground truth. Indeed, theoretically, observers could reach an almost perfect agreement (good reliability), but possibly different from the experts (low validity). As the ground truth consisted of observations that were previously noted during *in-person* monitoring and again confirmed by all three experts on video, validity in this investigation also reflected whether observers' assessments matched with the *in-person* recorded

clinical behaviors of the canines (with the exception of two observations that were observed by the experts but not available within the *in-person* observations). For optional observations, the term validity was not used as only two out of three experts identified the observation.

**2.6.1. Agreement between observers (reliability).** The Fleiss's kappa was used to assess the degree of agreement on the detection of the COs between the 23 observers after evaluating the same videos [28]. Fleiss's kappa ($\kappa$) is a generalization of Cohen's kappa for more than two observers with values ranging from –1 to +1. Table 3 shows the interpretation of the K values as suggested in literature.

The R package 'irr' (inter-rater reliability) was used to calculate the Fleiss's kappa between all observers across all COs separately on videos of primary and optional COs [29,30]. All analyses were performed using R Statistical Software v4.1.2 [29].

**2.6.2. Agreement between observers and experts on primary (ground truth, validity) and optional observations.** The statistical analysis was done for the primary and optional COs separately, first without including the comment section and thereafter with comments included. For each primary CO it was estimated how likely the observers would detect the ground truth as annotated by the three experts as follows. An indicator variable for a correct and wrong selection of a CO for a given video segment by a subject was defined:

$$Y_{ij} = \begin{cases} 1, & \text{if i}^{\text{th}} \text{ observer correctly identified the CO that matches the ground truth for j}^{\text{th}} \text{ video segment} \\ 0, & \text{otherwise} \end{cases}$$

for $i$ = 1, 2, …, I; $j$ = *1, 2, …, J*; with I = 23, J = 81.

The percentage of correctly detected COs was calculated for each video segment. The percentages correct for each CO were obtained by aggregating the percentages across all video segments and across all observers. The percentages were further aggregated across all COs to obtain an overall percent correct across all observers, COs and video segments using the formulae shown below.

The percentage of correctly detected primary COs per video segment across all observers for $j^{\text{th}}$ video segment equaled:

$$P_j = \frac{\sum_{i=1}^{I} Y_{ij}}{I} * 100, \quad \text{for } j = 1, …, J;$$

Similarly, the percentage correct for a given $g^{\text{th}}$ ground truth CO was calculated across all videos and observers:

$$p_g = \frac{\sum_{i=1}^{I} \sum_{j=1}^{J} CO_{gj} Y_{ij}}{I * Z_g} * 100, \quad \text{for } g = 1, \cdots, G \text{ with } G = 14,$$

**Table 3. Ranges of Fleiss's kappa and their respective interpretation for the agreement between different observers [28].**

| Kappa value | Meaning |
|---|---|
| K < 0 | Poor agreement |
| 0 ≤ K ≤ 0.2 | Slight agreement |
| 0.2 < K ≤ 0.4 | Fair agreement |
| 0.4 < K ≤ 0.6 | Moderate agreement |
| 0.6 < K ≤ 0.8 | Substantial agreement |
| 0.8 < K ≤ 1 | Almost perfect agreement |

where $CO_{gj} = \begin{cases} 1 \text{ , the } g^{th} \text{ ground truth CO is in video segment j,} \\ 0 \text{ , } otherwise \end{cases}$

and $Z_g = \sum_{j=1}^{J} CO_{gj}$ is the total number of video segments with $g^{th}$ ground truth CO.

Note that the number of video segments with $g^{th}$ ground truth CO are shown in <u>Table 2</u> and $\sum_{g=1}^{G} Z_g = 81$ , for the primary COs.

Finally, the overall percentage of correct detection of all COs as determined by the experts across all video segments was given by:

$$P = \frac{\sum_{i=1}^{I} \sum_{j=1}^{J} Y_{ij}}{I * J} * 100$$

A similar approach was used for the optional COs. For this analysis the total number of video segments was lower (20 instead of 81), meaning that J = 20 and G = 9 were used in all the above calculations for optional COs.

**2.6.3. Comparing experience levels and primary roles.** The agreement with the ground truth was calculated for the three experience levels of the observers using a mixed-effects regression model for binary data (correct/not correct). Similarly, the validity of the primary roles was statistically compared using the same statistical model. The model was fitted to the data assuming the observers as random factors. The experience level and primary role were separately included in the model as fixed effects to investigate their potential effects on the validity. A significant difference was declared when the p-value for a statistical test on the regression coefficient of the experience level or primary roles within the model was below 0.05 (p < 0.05).

## 3. Results

### 3.1. Concept

The goal of our current work was to investigate the interobserver reliability and construct validity to recognize COs in video recordings of preclinical safety studies in canines. The concept of defining the ground truth, evaluating the criterion validity, performing the video evaluations by the observers and the statistical data analysis is illustrated in <u>Fig 1</u>.

The experts' ratings showed a high criterion validity where only five out of 103 recorded in-life observations were not observed by ≥ two experts on the 70 videos (81 primary, 17 optional and five not observed). Therefore, expert ratings were considered the 'best approximate standardized test' to which observer ratings were compared in the evaluation of their construct validity.

Next, the interobserver reliability was investigated to evaluate how well the agreement was between the observers on identifying the 13 COs and normal behavior, without considering the experts' opinion. As the identification of COs is based on a subjective interpretation only, it was crucial to assess also the observers' construct validity by investigating their agreement with experts. This was done in a second part, for both the primary (ground truth) and optional observations; and focused on three of the four categories of Funder: *i) good trait:* whether some COs were more difficult to identify; *ii) good judge:* whether experience or primary role impacted the construct validity of the observations; and *iii) good target:* whether some videos were more difficult to judge. The effect of *good information* was not evaluated as all observers were given a refresher training prior to completing the questionnaire, and there were thus no 'untrained' observers. Construct validity is referred to as simply 'validity' throughout the manuscript.

Based on the initial validity results, a more in-depth analysis was performed on two parts. First on the identification of normal behavior: *i)* was there a correlation between missing actual COs and high scoring for normal behavior, and *ii)* did experience or primary role influence the recognition of normal behavior. Secondly, frequent mix-ups between certain COs were analyzed and it was investigated whether the presence of multiple primary COs complicated the observers' assessment.

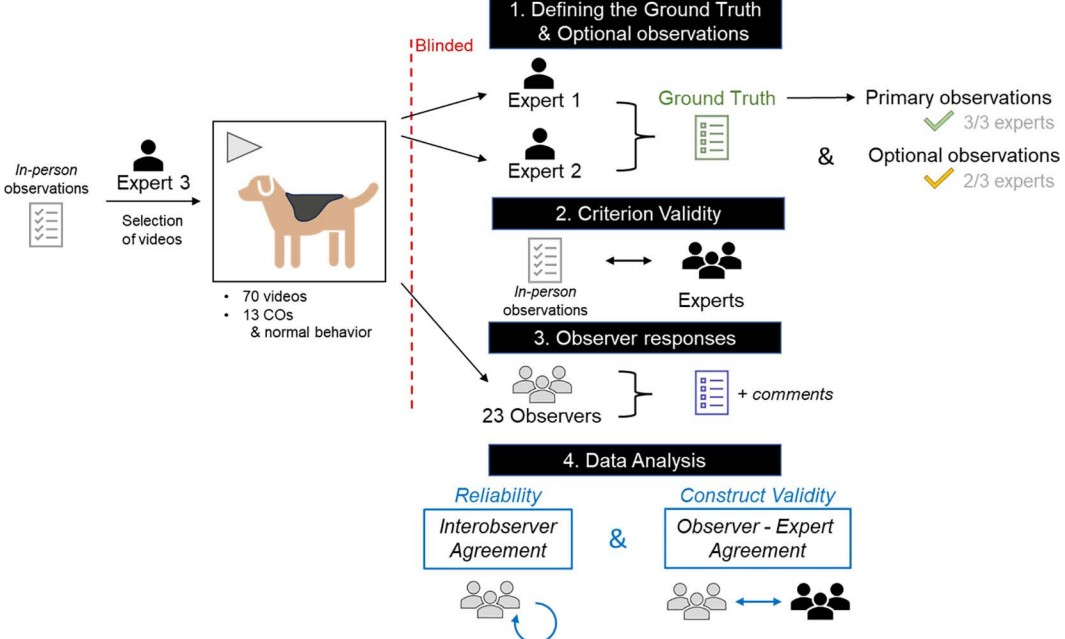

**Fig 1. Concept of this paper.** Seventy-three videos were initially selected by expert 3 based on in-person observations. All videos were subsequently blinded and scored by two other experts to compose the ground truth and optional COs. Three videos were rejected from the experiment due to lack of agreement. In a second step, the criterion validity of the experts' ratings was assessed. Finally, 23 observers evaluated the final set of 70 videos to determine the reliability and construct validity of the selected COs.

### 3.2. Agreement between observers (reliability)

The overall Fleiss's Kappa for the reliability was 0.70 indicating a substantial agreement between the 23 observers on all 15 possible selections (13 COs, normal behavior, and 'other, not listed'). Inclusion of comments from observers only slightly improved the overall Fleiss' Kappa to 0.72.

To identify the behaviors for which there was less agreement between the observers' assessments, the Fleiss's Kappa was calculated for each CO (Fig 2). A substantial to almost perfect agreement was reached for nine out of 13 COs, without incorporating the observers' comments. There was an almost perfect agreement on salivation, vomiting, aggressiveness, head shaking, circling and convulsions whereas a substantial agreement was reached for ataxia, limping and anxiety. Observers moderately agreed on tremors, excitation and normal behavior ('no clinical observation'). A fair agreement was reached for twitches and limb stiff/hypertonia. Inclusion of comments slightly improved the agreement on normal behavior (+0.07), ataxia (+0.05), limping (+0.05) and tremors (+0.04) (Fig 2).

### 3.3. Agreement between observers and experts (validity and *good trait*)

The substantial agreement between observers did not consider the ground truth. Hence, theoretically there could be a near perfect agreement (good reliability) between observers, although possibly different from the experts (low validity). Therefore, the next step was to investigate the validity for the primary observations.

The observers correctly identified 75% of the primary observations (S1 Fig), containing normal behavior and a total of 12 COs (n = 81 individual observations, see Table 2). The 75% validity score reflects a substantial agreement with the ground truth defined by all three experts. This shows that the observers' assessments were similarly reliable as valid. Inclusion of the comments slightly increased the overall validity to 77.5% with only two observers (V and J) showing a relevant increase in their score of +10% and +8%, respectively (S1 Fig).

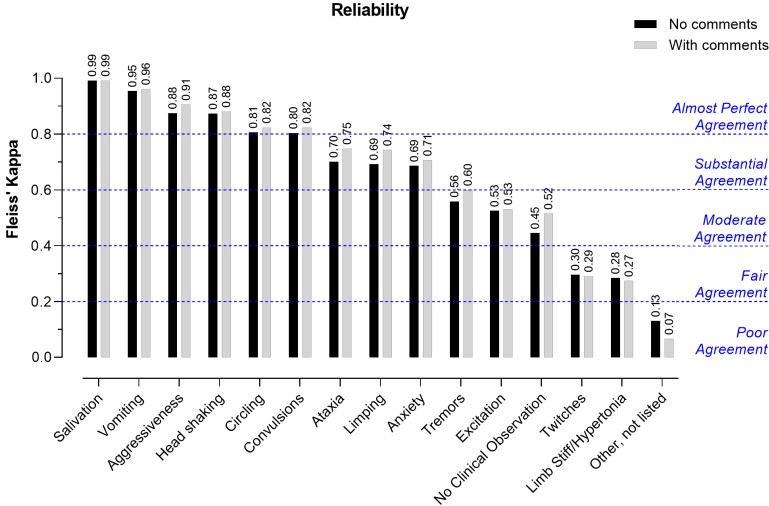

**Fig 2. Reliability for the 13 clinical observations and normal behavior.** The Fleiss' Kappa values are shown above each bar per COs and other possible selections (No clinical Observation or Other). The Fleiss' Kappa values are represented without (black bars) and with comment inclusion (grey bars). Ranges of Fleiss's kappa and their respective interpretation are indicated by dotted horizontal lines.

To investigate whether some observations were more difficult to recognize (*good trait*), the validity score was calculated for each primary observation separately (Fig 3A). A score of ≥ 90% was noted for six COs: salivation, aggressiveness, circling, vomiting, head shaking, and convulsions. There was a substantial validity for three COs with a score around 75%: limping, anxiety, and ataxia. For limb stiff/hypertonia and tremors, observers moderately agreed with the experts (50–70%) while for excitation there was a poor agreement (<50%). Interestingly, for normal behavior ('no clinical observation'), a moderate agreement was seen with a score of 56%. Inclusion of comments had no to minimal impact (≤+3%) for most behaviors, whereas a small impact was noted for ataxia (+5%) and "no clinical observations" (+6%).

For 11 out of 13 COs, the interpretation of the reliability and validity score was identical (Fig 3B). Only limb stiff/hypertonia and excitation showed slight differences between the reliability vs validity score. Limb stiff/hypertonia showed a poor reliability, below the threshold of >0.4 for Fleiss' Kappa which is often used as guide for minimum reliability [21,31]. When applying a similar arbitrary cut-off for the validity (>50%), excitation showed a poor validity (Fig 3B).

For only four out of 70 videos, the majority of observers agreed differently than experts. One video with tremors were scored as twitches instead; and three videos of normal behavior were detected as a specific CO (limb stiff/hypertonia, excitation and circling).

The agreement between observers and experts was also investigated for the optional observations which were expected to be more challenging to identify as only two out of three experts found agreement. In contrast to the overall validity of 75% for the primary observations, the overall agreement for the optional COs (n = 17, Table 2) was much lower with a score of 21.7% between all observers and two experts. The effect of comment incorporation was minimal (+1.1%). The individual scores for the optional observations resulted in poor agreement (<50%). The highest scoring optional observation (twitches, 36%) still scored much lower than the lowest scoring primary observation (excitation, 47%) (S2 Fig).

### 3.4. Effect of difficulty level of the videos on CO recognition (*good target*)

To evaluate Funders' *good target*, experts 1 and 2 scored the degree of difficulty to detect each of the 81 primary and 17 optional COs on video. The specific CO or normal behavior was scored as *easy* or *difficult* to recognize, whereas "*Not*" reflected COs not confirmed by one of the experts (*i.e.,* for optional COs). Primary COs that were marked as *easy* by both

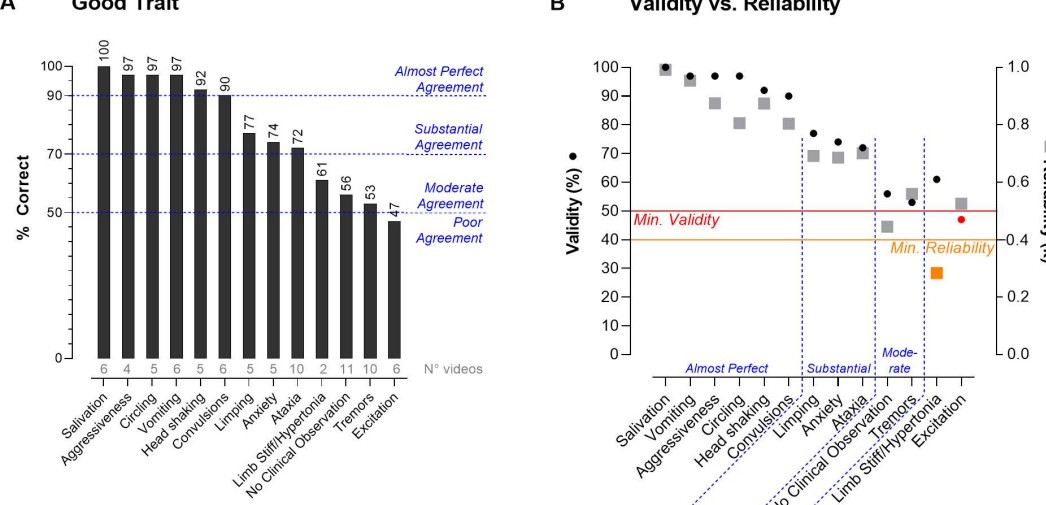

**Fig 3. Validity for all 12 primary COs and normal behavior: good trait and comparison with reliability.** (A) The average % correct score for each CO is depicted above each bar; the number of videos containing the respective primary CO is represented in grey below the bars. Data represents results without including the observers' comments. (B) Validity (circles) is plotted in % agreement to the ground truth, for reliability (squares) the Fleiss's kappa values are plotted. Arbitrary cut-offs for minimal agreement are marked by the red line for validity and by the orange line for reliability. Data points that don't meet the minimal cut-offs are marked in red (validity) and orange (reliability). The different categories of agreement are shown in blue.

experts (e.g., easy/easy) were also easily detected by the observers with 82.8% correct. Primary observations that were marked as *difficult* by one or both experts were harder to identify and showed scores of ≈60%. Finally, optional observations showed the lowest scores of ≈18–27% (Fig 4A).

Presence of multiple primary COs within a video complicated the experts' and observers' assessment. For the experts, the detection of each primary CO did not seem to be affected when they occurred simultaneously. Indeed, both blinded experts found it *easy* to detect all simultaneously occurring primary COs in five of the 10 videos. However, for the observers, it was more difficult to recognize multiple individual primary COs, since in all 10 videos there was clearly one CO with a higher validity compared to other primary CO(s) within that video (S1 Table).

### 3.5. Effect of experience level and primary role on validity (*good judge*)

The group of observers consisted of three-experience levels: five participants had minimal experience (22%), 10 had moderate experience (43%), and eight had high experience (35%) in behavioral assessment of canines. When it came to the observers' main functional role, there were eight lab technicians (35%), nine scientists (39%) and six veterinarians (26%).

There was no statistical difference between the validity score of the three different experience levels and primary roles, however a slight trend towards a higher validity was observed for more experienced observers (Fig 4B). Similarly, technicians tended to have a higher-than average score compared to other roles (Fig 4B).

For the optional observations, all expertise levels had similar scores (moderate 22.4%, high 21.3% and minimal 21.2%); and technicians and veterinarians scored slightly better than scientists (22.8% and 23.4% versus 19.6%, respectively) (data not shown).

### 3.6. Effect of experience level and primary role on identifying normal behavior

As normal behavior (i.e., 'No Clinical Observation') proved challenging to correctly identify with a score of 56% (Fig 3A), it was investigated *i)* whether observers with a high score for detecting normal behavior (true negative) would miss more

**A   Good Target**

**B   Good Judge**

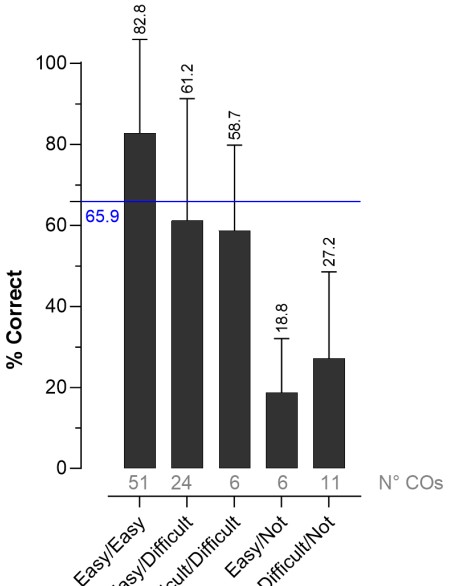
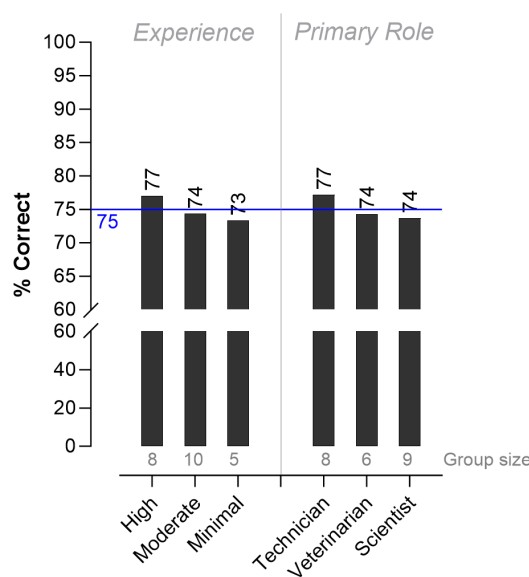

**Fig 4. *Good target* and *good judge*.** (A) Comparison of observers' scores between difficulty levels of detection as assessed by both blinded experts (*good target*). Black bars represent the average % correct without incorporating the comments. Data is represented as mean ± St Dev. The horizontal blue line represents the average overall score across all difficulties. (B) Comparison of the overall validity based on experience and functional role (*good judge*). Black bars represent the data without comment inclusion. The overall % correct for each group separately is depicted above each bar. The horizontal blue line represents the overall average validity across all three groups.

actual COs (false negative); and *ii)* whether experience or primary role would have an influence on detecting normal behavior and/or on the number of actual COs that were missed.

In general, there was a broad range (27–91%) in correctly identified normal behavior (true negatives) based on 11 video snippets (Fig 5A). The proportion of missed COs (false negatives) remained limited, ranging from 0–5% out of 59 video snippets (excluding a single outlier of 10%) (FigA). The five observers with a high true negative score (≥ 82%) showed higher false negative scores (3–10%); while 50% of the observers with a lower true negative score (≤ 73%) did not miss any CO. These results point towards an observers' bias not to miss any CO; and indicate that observers with a high true negative score were possibly more prone to miss COs by scoring those as normal behavior (higher false negative score) (Fig 5A).

All levels of experience were comparable in correctly identifying normal behavior with averages ranging from 54.5% to 58.2% (Fig 5B). However, highly experienced observers tended to miss fewer COs than colleagues with minimal experience. Their average false negative score was 1.5% compared to 2.5% and 3.4% for moderate and minimal experience, respectively (Fig 5B).

Technicians seemed more performant in correctly recognizing normal behavior with an average score of 62.5% compared to scientists (53.5%) and veterinarians (51.5%). The latter appeared to have a lower false negative rate of 1.1% compared to technicians (2.3%) and scientists (3.2%). On individual level though, it was remarkable that 62.5% of the technicians and 50% of the vets succeeded in detecting abnormal behavior (0% false negatives); while only 11.1% of the scientists accomplished the same (Fig 5C).

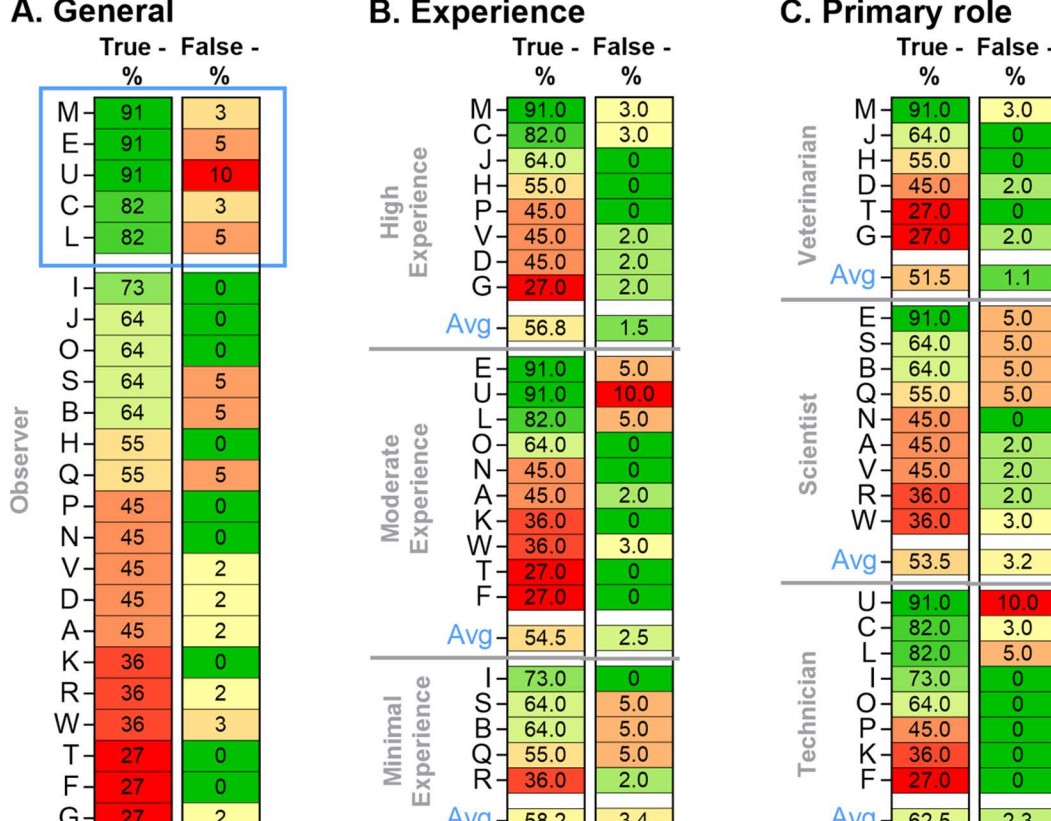

**Fig 5. Heatmaps showing true negative and false negative rates for detection of normal behavior.** (A) True and false negative rates for each observer separately, ranked from high to low true negative rate. The blue box depicts the observers with a high true negative score (≥ 82%). (B) True and false negative rates for different experience levels. The blue 'Avg' value depicts the average rates per experience level. (C) True and false negative rates for the different primary roles. The blue 'Avg' value depicts the average rates per functional role.

### 3.7. Challenges in identifying and differentiating COs

Detailed analysis showed that certain COs were swapped frequently: tremors vs. twitches, limping vs. limb stiff and ataxia vs. limb stiff/limping. These COs were therefore investigated more in depth to evaluate *i)* whether they were in fact distinguishable based on video footage alone, and *ii)* whether there was a difference in detection if the CO was present as the only primary CO compared to when also other primary COs were present. For excitation, this last aspect was also evaluated, as this was the CO with the lowest validity.

For tremors, 10 videos showed tremors as primary observation; in four of those, other primary observations were also present. Observers could more easily detect the tremors in case they were the only primary observation: the overall average score of 54% increased to 63% for that subset of videos (S2 Table). Based on video footage alone, it proved difficult for most observers to differentiate between tremors and twitches: a number of observers identified twitches instead of tremors in six out of eight videos. If both COs were considered correct, the overall score improved from 54% to 65% for all videos with tremors as primary observation, and from 63% to 83% for videos in which tremors were the only primary observation (S2 Table).

Limping was present in five videos, whereas limb stiff was present in two other videos as a primary observation. In general, limping was easier to detect than limb stiff, with an overall average score of 76% vs. 61%, respectively (S3 Table).

Limping as a primary observation was often confused with limb stiff with an average score improving from 76% to 90% when both COs were considered correct (S3 Table).

Ataxia was present 10 videos as primary observation; four of those videos also showed other primary COs. Observers were generally able to detect ataxia with an overall score of 72%, while ataxia as only primary CO had a score of 85% (S4 Table). Ataxia was occasionally confused with limb stiff, and rarely with limping, although this did not substantially influence the overall score (+5%) (S4 Table).

Finally, excitation was present in two videos as the only primary observation, for which the validity was very high (96% and 100%). This was in sharp contrast to the significantly lower validity (13%−35%) for the four remaining videos in which also other (more obvious) primary COs were identified (circling and anxiety).

The importance of clear definitions in what to state as separate observations is also highlighted in case of simultaneous occurrence of COs. Some observers for instance considered excitation being associated with circling, while both observations should be stated.

## 4. Discussion

*In vivo* preclinical safety studies in canines, amongst other species, are critical in drug development to determine the potentially adverse effects of a novel drug candidate. Video surveillance is a valuable tool to reduce the risk of missing clinical observations or adverse behavior and to enable a more objective and quantifiable monitoring without the need for human interference. The current research aimed to understand how well observers were able to recognize specific clinical behaviors in canines on video, by evaluating the reliability (agreement between observers) and construct validity (agreement between observers and experts) of the observers' assessments. The ground truth consisted almost exclusively of observations that were previously recorded during *in-person* monitoring which were confirmed by all three experts on video (high criterion validity). Therefore, (construct) validity in this investigation reflected whether observers' and experts' assessments matched, but also whether observers were able to detect clinical behaviors that were noted during *in-person* monitoring.

Given the substantial overall agreement between all observers, and more importantly between the observers and the experts, we conclude that video data permits reliable and valid identification of clinical canine behavior by trained observers. Similarly, Lazarowski *et al.* showed a substantial to almost perfect agreement between live and video scorers on three behavioral tests for selecting puppies for future service duties. On the other hand, they demonstrated a low agreement for the subjective overall assessment of the dogs' performance [15]. Like our research, this also points out that there are limitations to video assessment to keep in mind, related to Funders' categories *good trait*, *good judge* and *good target* [7,10].

G*ood trait* refers to some behaviors being more difficult to detect than others, *i.e.,* having a lower reliability and validity. This was pointed out by a number of publications on canine behavior [7,9,12,13,15,17,18,20–22], but this investigation is the first to show this for drug-induced deviant canine behavior within pharmaceutical research. As suggested in a number of publications on a variety of animal species [7,26,32,33], this research also indicates that a good interobserver and ground truth agreement is more difficult to attain for subtle behaviors, *e.g.,* with less available or less clear visual cues to be interpreted by the observers. For the majority of the COs, the results for reliability and validity were indeed very comparable. Salivation, aggressiveness, circling, vomiting, head shaking and convulsions are all associated with clear visual cues (*e.g.,* saliva dropping, appearance of vomit, sudden change in muscle tonus) and were considered easy to identify as they showed an almost perfect reliability and validity. On the other side of the spectrum, limb stiff/hypertonia and excitation were the most subtle behaviors when judging on video. This reflected in slight differences between reliability and validity, with limb stiff showing a higher validity and excitation showing a higher reliability. While our results show that the large majority of clinical behavior can be identified with good reproducibility and validity using video footage, certain COs are challenging to distinguish as this is based on subtle differences that might not be clearly visible on video recordings [7,9], *i.e.,* tremors vs. twitches and limping vs. limb stiff/hypertonia. In addition, not all phenotypes of 'easy to

identify' observations are clearly recognizable, as is illustrated by the much lower validity when these COs were marked as optional.

To the best of our knowledge, our research is the first to present data on the recognition of clinical canine behavior. Hence, not many comparisons can be made to behaviors from other canine publications, apart from aggression and anxiety. Aggression was easily recognized in our work, as also Correia *et al* pointed out [34]. However, in the publication of Tami *et al.* canine-to-canine aggression was not well recognized, not even by experienced owners and trainers. During their research, contact between canines was avoided for safety reasons which might have impaired the observers' ability for a correct interpretation [18]. Our substantial reliability and validity for anxiety was comparable to results of other publications in which anxiety in canine-to-canine interactions (67% correct) [18] and an anxious response to a vacuum cleaner or sudden noise [26] could reliably be recognized with intra-class correlation coefficients (ICCs) of 0.89 and 0.94, respectively.

For normal behavior, there was a moderate reliability and validity, inherent to the exercise of identifying all clinical behaviors. The fear of missing COs translated in an observers' bias, making them more prone to score potential COs instead of identifying normal behavior. Indeed, there was a wide range in correctly identified normal behavior, while only very few COs were missed – most of them by observers who were more performant in identifying normal behavior (Fig 5). An additional explanation for the moderate agreement on normal behavior was the short length of the video snippets, making it more challenging to determine whether the behavior was still normal or not. In case of *in-person* monitoring, observers would monitor the behavior of the animal for a longer period of time and compare it to baseline conditions in order to make the correct decision. In addition, the personal interaction with the animal is a great aid to determine when a behavior is abnormal. This observers' doubt in judging normal behavior from the video snippets is indeed reflected in the noteworthy impact of comment inclusion on both the reliability (+0.07, Fig 2) and validity (+6%).

The second point of focus was on *good judge, i.e.,* whether some people were better observers than others. In this investigation, there was no difference in the validity of the assessments between different experience levels for both primary and optional observations. However, the highly experienced group missed fewer actual observations than their less experienced colleagues. Our findings are in line with a number of publications on canines showing that novice observers can obtain satisfactory agreement with experts [7,18,26], but that this can be improved by gaining experience and following training [7,9,22]. Indeed, the substantial validity regardless of experience level in our research may be explained by the basic experience level of all observers in combination with the refresher training that was given on the selected COs prior to assessing the videos. This illustrates the importance of setting clear definitions to reach a good agreement. Also for the primary roles, there was no statistical difference, although technicians tended to have a higher-than-average validity, and a similar trend was seen for optional observations. Technicians seemed more performant in correctly differentiating between normal behavior and COs (Fig 5C). This was expected as they are responsible for the *in-person* monitoring of animals and routinely use video surveillance for observation. Interestingly, veterinarians missed on average the fewest COs, whereas scientists missed the most COs.

Thirdly, this research demonstrated that *good target* also played a role. Indeed, irrespective of the observation itself, the clinical behavior in some videos was more difficult to identify than that in others. As our cameras are placed in fixed positions, important visual cues (*i.e.,* the animal itself, or vomit produced) were sometimes occluded by for example the food hopper or bin enrichment. This limitation of video recordings was also pointed out by Fratkin *et al.* [26]. Other than occlusion, the CO in that video can also be a more subtle or less clear expression of the behavior, making it more difficult to identify.

A limitation to this research is that there were only few (or no) videos for limb stiff/hypertonia and twitches as primary CO, which restricts the conclusions that could be made for these COs. A second limitation which was received as feedback from the majority of the observers, was that certain videos were too short to enable a good interpretation, in particular for those more subtle behaviors. Another important remark is that some videos contained multiple primary COs

which impacted both *good trait* and *good target.* Indeed, the observers focused on the most obvious CO which indirectly increased the difficulty for detecting more subtle COs; and/or a certain CO was considered as part of another more dominant CO and was therefore not separately scored.

In summary, video recordings are a valuable addition to in-person monitoring as they allow to re-watch certain behaviors and COs for more detailed evaluation of the animals; and monitor the animals without human presence. Video recordings can be used instead of direct observation when the latter is not feasible (e.g., outside working hours) for the majority of the clinical observations, keeping in mind that subtle observations or subtle phenotypes of a certain observation might become less clear and can therefore be missed. This could be circumvented by having multiple observers assessing the videos and possibly grouping certain similar observations together, *e.g.,* tremors and twitches.

Overall, this work illustrates the important role of video-analysis in preclinical studies to enable a full coverage of the animal's activity and behavior similar to [35,36], and to decrease the risk of missing critical adverse events. The availability of video surveillance also fulfills an important aspect on 3Rs, more specifically on the refinement and monitoring of animal health and wellbeing. One of the major downsides of manual video analysis is the very time-consuming work making 24/7 coverage unrealistic to reach. The results of our research encourage using video footage for artificial intelligence-based approaches (*i.e.,* computer vision) to develop video-based continuous monitoring of canines. In the near future, this might lead to automatic detection and quantification of animal activity and clinical behavior.

## 5. Conclusion

Our results demonstrated that clinical behavior in canines can be detected with good reliability and validity on video footage, irrespective of the observers' experience level and functional role. However, there are limitations to keep in mind. Subtle visual cues can become less clear, which not only increases the difficulty of distinguishing between similar COs, but also makes it more challenging to detect subtle behaviors associated with these cues. From all 12 primary COs and normal behavior, only limb stiff/hypertonia and excitation had a poor validity or reliability. Altogether, we strongly believe that video surveillance should become part of the future monitoring of *in vivo* preclinical safety studies, not only to increase our understanding of the safety profile of drug candidates, but also to further improve the refinement of those studies as part of the 3Rs. Our results will serve as the first critical step in exploring the use of computer vision AI to enable continuous 24/7 monitoring in canine studies.

## Supporting information

**S1 Fig. Overall validity per observer.** Observers are ranked from high to low validity scores, without (black) and with (grey) comment incorporation. The blue lines represent the overall average validity score across all observers.
(TIF)

**S2 Fig. Agreement between observers and experts on the nine optional observations.** The average % correct score for each optional CO is depicted above each bar; the number of videos containing the respective optional CO is represented in grey below the bars. Data represents results without including comments.
(TIF)

**S1 Table. Detection of simultaneously occurring primary COs.** For each CO separately in each video, the validity and expert scoring are show. Primary CO 1 reflects the CO with the highest validity, primary COs 2 and 3 reflect the more subtle COs with a lower validity. The highlighted cells (grey) mark the five videos in which all simultaneously occurring COs were marked as easy by both experts.
(TIF)

**S2 Table. Detection of tremors and differentiation with twitches.** Blue values reflect the % of observers that correctly identified the tremors as tremors or as a combination of tremors and twitches, dotted cells are % of observers that scored

the tremors as twitches, green values are the % of observers that identified tremors and/or twitches. White cells are the videos in which tremors were present as only primary CO, light grey cells reflect videos in which also other primary COs were present. Videos A10 and A28 were excluded from the analysis as twitches were scored as optional observation by two experts in those videos and they were also detected by a number of observers.
(TIF)

**S3 Table. Differentiation between limping and limb stiff.** Blue values reflect the % of observers that correctly identified limping (four videos) or limb stiff (two videos) or both, dotted cells are % of observers that wrongly scored limping as limb stiff or limb stiff as limping, green values are the % of observers that identified limping and/or limb stiff. White cells are the videos in which limping or limb stiff were present as only primary CO, light grey cells reflect the limping video in which also another primary COs was present. Limping in video A47 was not registered during *in-person* monitoring.
(TIF)

**S4 Table. Detection of ataxia and differentiation with limping/limb stiff.** Blue values reflect the % of observers that correctly identified ataxia, dotted cells are % of observers that scored ataxia as limping/limb stiff, green values are the % of observers that identified ataxia and/or limping/limb stiff. White cells are the videos in which ataxia was the only primary CO, light grey cells reflect the videos in which also other primary COs were present. Video A53 was excluded from the analysis as limb stiff was scored as optional observation by two experts and it was also identified by several observers.
(TIF)

**S1 Dataset. Excel file containing the experts' observations, their scoring and all 23 blinded and completed questionnaires.**
(XLSX)

**S1 File. File containing all relevant statistical results.**
(PDF)

## Acknowledgments

The authors wish to thank all 23 observers for their interest and vigorous participation in this research, and J. Wilson and Y. Van Bekkum for reviewing this manuscript.

## Author contributions

**Conceptualization:** Fetene Tekle, Greet Teuns, Jill Witters, Ivan Kopljar.

**Data curation:** Eline Eberhardt, Fetene Tekle.

**Formal analysis:** Eline Eberhardt, Fetene Tekle.

**Funding acquisition:** Ivan Kopljar.

**Investigation:** Eline Eberhardt, Greet Teuns, Jill Witters, Bianca Feyen, Sarah De Landtsheer, Ivan Kopljar.

**Methodology:** Fetene Tekle, Greet Teuns, Jill Witters, Ivan Kopljar.

**Project administration:** Eline Eberhardt, Ivan Kopljar.

**Resources:** Bianca Feyen, Sarah De Landtsheer.

**Software:** Fetene Tekle.

**Supervision:** Ivan Kopljar.

**Visualization:** Eline Eberhardt.

**Writing – original draft:** Eline Eberhardt.

**Writing – review & editing:** Eline Eberhardt, Fetene Tekle, Greet Teuns, Bianca Feyen, Sarah De Landtsheer, Ivan Kopljar.

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
