## [Decision Letter · Decision Letter 0]

PONE-D-24-24225Application of video surveillance in preclinical safety studies in Canines: understanding the interobserver reliability and validity to recognize clinical behavior.PLOS ONE

Dear Dr. Kopljar,

Thank you for submitting your manuscript to PLOS ONE. After careful consideration, we feel that it has merit but does not fully meet PLOS ONE’s publication criteria as it currently stands. Therefore, we invite you to submit a revised version of the manuscript that addresses the points raised during the review process.

Several of the authors had significant reservations about the experimental design. It is reasonable to withdraw this manuscript and resubmit in the future if you need additional time to resolve these issues.PLOS Data policy also requires authors make all data underlying the findings described in their manuscript fully available without restriction. If you are not able to adhere to that policy due to privacy concerns, you should consider an alternative journal. 

We look forward to receiving your revised manuscript.

Kind regards,

Cord M. Brundage, D.V.M., Ph.D.

Academic Editor

PLOS ONE

Journal Requirements:

“AlI have read the journal's policy and the authors of this manuscript have the following competing interests: all the authors were employed by company Janssen Research and Development, Janssen Pharmaceutical Companies of Johnson & Johnson.”

4. We note that Figure 1 in your submission contain copyrighted images. All PLOS content is published under the Creative Commons Attribution License (CC BY 4.0), which means that the manuscript, images, and Supporting Information files will be freely available online, and any third party is permitted to access, download, copy, distribute, and use these materials in any way, even commercially, with proper attribution. For more information, see our copyright guidelines: http://journals.plos.org/plosone/s/licenses-and-copyright.

Additional Editor Comments :

Please make sure you address reviewer concerns both in your response to reviewers and in the manuscript or indicate why it would inappropriate to do so.

Reviewers' comments:

Reviewer's Responses to Questions

**Comments to the Author**

1. Is the manuscript technically sound, and do the data support the conclusions?

Reviewer #1: Yes

Reviewer #2: No

Reviewer #3: No

Reviewer #4: Yes

2. Has the statistical analysis been performed appropriately and rigorously? 

Reviewer #1: Yes

Reviewer #2: No

Reviewer #3: Yes

Reviewer #4: Yes

3. Have the authors made all data underlying the findings in their manuscript fully available?

Reviewer #1: Yes

Reviewer #2: No

Reviewer #3: No

Reviewer #4: Yes

4. Is the manuscript presented in an intelligible fashion and written in standard English?

Reviewer #1: Yes

Reviewer #2: Yes

Reviewer #3: Yes

Reviewer #4: Yes

5. Review Comments to the Author

Reviewer #1: This manuscript looks at the reliability and validity of using video recordings to assess clinical observations, taking into account observer experience level and role. Overall I think the manuscript is well written and clearly describes the methods and results, while identifying the limitations of video observations, but providing thoughtful suggestions on using such measures to train machine learning technologies for automated detection of behavior. I have a few minor comments/questions below.

Materials and methods:

L127: How were the video snippets selected and by whom?

L130: What types of studies? Were the animals always in the same housing set up?

L136: How long were animals separated after feeding?

L140: How were the videos blinded and by whom?

L152-153: Why were twitches only an optional observation?

L184: Were the videos randomized in the same order for all observers or randomized per observer? How were videos randomized?

Results:

L273: Error! Reference source not found… - This error is found in numerous places throughout the manuscript.

Reviewer #2: The experimental design of the study has some important to note flaws due to the presence of only 2 experts for generating the ground truth for comparison with the observers’ ratings of the videos. Since the ground truth only includes the observations that both experts share, this is essentially sub-selecting only the easier to interpret aspects of the clinical observations and eliminating the role of less easily observed aspects of the animal behavior. This leads the conclusions of the study to be flawed as well since the authors are maintaining that non-experts can interpret as well as experts. However, the authors are only evaluating their correctness based on easier to interpret observations (only those that are shared by 2 experts).

To address these issues, I would recommend major revision of the study design by generating the ground truth based on the observations of more than 2 experts. The ground truth should be observations that show at least a majority prevalence in a sample of exerts to be reflective of the population of exerts instead of only having the study design catered to the observations that overlap between only 2 observers. This automatically eliminates from consideration of the study more difficult to observe behaviors. This leads to the author’s conclusions that the video observations are as accurate as the expert observes overall. If the video observers were assessing behaviors that were more difficult for experts to observe, then it is likely that the accuracy of the video observers would be reduced overall when they are assessing more nuanced behaviors. The current study design introduces potential error if one of the experts makes an error, which could be resolved if more than 2 experts are used to establish the ground truth. The authors should include in their study design justification a power analysis that indicates the rationale for choosing the number of experts given their ability to establish a reliable conclusion. The experts should have a greater inter-expert reliability index than non-experts if they are truly an expert in the subject matter. The study did not conduct this type of analysis for more than 2 subjects to determine what number of experts would contribute to a sufficient baseline.

For example, the authors should reference studies in a range of fields including animal behavior and ethology (Anderson et al 2014; Bohnslav et al 2021) and other fields such as medical image annotation (Athanasiou et al 2023) and radiology (Lebovitz et al 2021), for standards on the number of experts that should be used for establishing the ground truth as these studies use more experts for their study design, ranging from 3 to 23. Since studies in animal ethology have noted that 2 human expert observers only agree around 70%, it would be necessary to obtain a dataset of multiple observers (at least 3-5 experts) so that an observation can be determined to be present in most expert assessments. These standards for study design are especially important given that the authors are making conclusions regarding the relevance of their findings for the area of computational animal behavior, which would require a larger number of experts as a standard for training models than they used in this study.

The authors state that their data is not fully available due to restrictions. This does not meet the PLOS One journal requirement for data availability in the current state. The UNESCO Recommendations on Open Science state that restrictions “are only justifiable on the basis of the protection of human rights, national security, confidentiality, the right to privacy and respect for human subjects of study, legal process and public order, the protection of intellectual property rights, personal information, sacred and secret indigenous knowledge, and rare, threatened or endangered species” and that “In cases where data cannot be openly accessible, it is important to develop tools and protocols for pseudonymizing and anonymizing data, as well as systems for mediated access, so that as much data as possible can be shared as appropriate.”. The authors should address these points regarding restriction of their data, and if they are not applicable, then they should provide access to their data in a public repository such as Open Science Framework.

Overall, if these major concerns are addressed, this would lead to a scientifically rigorous approach as the study goals are worthwhile to examine. As it stands, the issue with using only 2 experts for the study leads to results that are obscured by the elimination of too much data from consideration to be reflective of whether video observers can be as reliable and as valid as experts. The author’s study purpose for evaluating the effectiveness of video analysis is important to extend the ability to acquire data from animal studies, so a more rigorous approach would be important to achieve results that support these goals.

Reviewer #3: The current manuscript presents a study assessing the validity and reliability of measures used to evaluate behavioural responses in lab dogs.

The manuscript is well written and clear. However, what authors are describing in this manuscript as original research, is a common part of the standard early steps for measuring behaviour (e.g., ethogram building, observer training and reliability assessment; see Bateson and Martin 2021), with the add on of testing whether level of experience (subjectively and self assessed) has an effect on reliability.

Indeed, the validity and reliability of behavioural measurements in safety assessment studies are quite relevant and needed. Behavioural indicators of clinical concerns in laboratory animals are often missed or unreliably scored; all of which can decrease translatability, the quality of data and compromise the welfare of animals. However, I fail to see how results from the current manuscript contribute to address this issue:

1. The study lacks an independent assessment of the validity of the measures included – video scoring performed by two experienced observers within the same department, does not validate a behavioural/clinical measure. The comparison between experts scoring of videos and the rest of observers is still an indicator of consistency and measures can be reliable/consistent but have low validity – simply put, to which degree a measure is reflective of what it is intended to measure (the phenomenon of interest; e.g., the animal’s clinical or emotional state or the adverse effects of a compound; see Nunnally 1978, for a reading on face and construct validity).

2. Definitions of selected clinical observations confound behaviours with the inferences that can be made regarding the animals’ internal emotional and clinical states, and in several cases lack a clear description of objectively measurable behaviours. I understand that these definitions follow an internal glossary, thus are practical and applicable for their research lab, but reduces the relevance of its findings outside their group.

4. Videos were selected and cut for the purpose of this study. I question whether these videos are representative of the variation (in quality and behavioural diversity) seen in real post dosing cage side observation videos; hence, reliability assessed in these videos may not be translatable to actual video scoring for assessing drug adverse effects.

5. It does not address the issue of how video recordings can be used instead of direct observation.

Minor: several references marked as: Error! Reference source not found..

Reviewer #4: 1. Rewrite lines 91-92 for better understanding.

2. Rewrite lines 107-110 for clear understanding.

3. Could you please explain why there is a huge difference in the time range of the video snippets in the lines 127-128.

4. How many dogs were observed in each video please mention in the line 140.

6. PLOS authors have the option to publish the peer review history of their article (what does this mean? ). If published, this will include your full peer review and any attached files.

**Do you want your identity to be public for this peer review?** For information about this choice, including consent withdrawal, please see our Privacy Policy .

Reviewer #1: No

Reviewer #2: No

Reviewer #3: No

Reviewer #4: No

---

## [Author Response · Author response to Decision Letter 1]

21 Feb 2025

A rebuttal letter has been uploaded as a separate file labeled 'Response to Reviewers' (as requested in the Decision letter)

---

## [Decision Letter · Decision Letter 1]

PONE-D-24-24225R1Application of video surveillance in preclinical safety studies in Canines: understanding the interobserver reliability and validity to recognize clinical behavior.PLOS ONE

Dear Dr. Kopljar,

Thank you for submitting your manuscript to PLOS ONE. After careful consideration, we feel that it has merit but does not fully meet PLOS ONE’s publication criteria as it currently stands. Therefore, we invite you to submit a revised version of the manuscript that addresses the points raised during the review process.

We look forward to receiving your revised manuscript.

Kind regards,

Cord M. Brundage, D.V.M., Ph.D.

Academic Editor

PLOS ONE

Additional Editor Comments :

Please note that PlosOne requires that your article is free of all copyediting and typing errors before acceptance.

Reviewers' comments:

Reviewer's Responses to Questions

**Comments to the Author**

1. If the authors have adequately addressed your comments raised in a previous round of review and you feel that this manuscript is now acceptable for publication, you may indicate that here to bypass the “Comments to the Author” section, enter your conflict of interest statement in the “Confidential to Editor” section, and submit your "Accept" recommendation.

Reviewer #1: All comments have been addressed

Reviewer #2: (No Response)

Reviewer #3: (No Response)

Reviewer #4: All comments have been addressed

2. Is the manuscript technically sound, and do the data support the conclusions?

Reviewer #1: Yes

Reviewer #2: Yes

Reviewer #3: Partly

Reviewer #4: Yes

3. Has the statistical analysis been performed appropriately and rigorously? 

Reviewer #1: Yes

Reviewer #2: Yes

Reviewer #3: Yes

Reviewer #4: Yes

4. Have the authors made all data underlying the findings in their manuscript fully available?

Reviewer #1: Yes

Reviewer #2: Yes

Reviewer #3: Yes

Reviewer #4: Yes

5. Is the manuscript presented in an intelligible fashion and written in standard English?

Reviewer #1: Yes

Reviewer #2: Yes

Reviewer #3: Yes

Reviewer #4: Yes

6. Review Comments to the Author

Reviewer #1: The authors have addressed all my previous comments. I have no further comments to provide.

Reviewer #2: The authors have addressed my primary concerns of the number of experts described for establishing the necessary and optional observations for COs as well as the availability of raw data in the manuscript.

There are some remaining minor editorial changes that should be made, including:

Line 161: “When all 2 experts agreed…” should read “When all 3 experts agreed…”

Authors should make sure that they are consistent between the use of numerals and spelled out number notation ie. Three vs 3 and two vs 2 should remain consistent throughout the manuscript. This is an issue throughout the sections of the manuscript, so I will not note each case of inconsistency – the authors should thoroughly review the manuscript to maintain consistent notation. Typically, numbers are spelled out to represent numbers one through nine.

Reviewer #3: Authors have not yet satisfactorily addressed my first issue - which is non-trivial as it represents part of the core of the manuscript. In fact, this was also brought up by reviewer 1 and your response also did not address his comment. Having 3 instead of 2 experts does not “fix” the issue, their outcome is still a reliability measure rather than an evaluation of validity. Please provide a thorough explanation of the type of validity evaluated.

Reviewer #4: (No Response)

7. PLOS authors have the option to publish the peer review history of their article (what does this mean? ). If published, this will include your full peer review and any attached files.

**Do you want your identity to be public for this peer review?** For information about this choice, including consent withdrawal, please see our Privacy Policy .

Reviewer #1: No

Reviewer #2: No

Reviewer #3: No

Reviewer #4: No

---

## [Author Response · Author response to Decision Letter 2]

23 May 2025

Rebuttal Letter

Thank you for your valuable comments. We have addressed all the comments. See our answers in blue italic.

Reviewer #2: The authors have addressed my primary concerns of the number of experts described for establishing the necessary and optional observations for COs as well as the availability of raw data in the manuscript.

There are some remaining minor editorial changes that should be made, including:

Line 161: “When all 2 experts agreed…” should read “When all 3 experts agreed…”

Authors should make sure that they are consistent between the use of numerals and spelled out number notation ie. Three vs 3 and two vs 2 should remain consistent throughout the manuscript. This is an issue throughout the sections of the manuscript, so I will not note each case of inconsistency – the authors should thoroughly review the manuscript to maintain consistent notation. Typically, numbers are spelled out to represent numbers one through nine.

Thank you for pointing out the inconsistencies. We thoroughly reviewed the manuscript again to correct them. All the numbers are spelled out now, except when referring to expert 1, 2 or 3. These are kept as numerical for clarity.

Reviewer #3: Authors have not yet satisfactorily addressed my first issue - which is non-trivial as it represents part of the core of the manuscript. In fact, this was also brought up by reviewer 1 and your response also did not address his comment. Having 3 instead of 2 experts does not “fix” the issue, their outcome is still a reliability measure rather than an evaluation of validity. Please provide a thorough explanation of the type of validity evaluated.

We included in the manuscript a thorough explanation of the types of validities that were evaluated (Materials & Methods section 3, Results section 1 and updated Fig 1) and explain further below.

Our main purpose was to investigate whether observations made in-life can be correctly recognized on video footage. Therefore, we first evaluated the criterion validity between the in-life observations and experts on video and secondly the construct validity between the experts and observers.

For the criterion validity, we investigated whether experts saw the same observations on video as those registered during the in-life phase of the studies. These in-life observations are considered as the current ‘gold standard’ as they are based on an internal lexicon which is well aligned between pharma companies and CRO’s and on which all relevant personnel is trained.

To cite Rebecca K Meagher on criterion validity in animal welfare research “to assess something that can be measured directly in some other way by comparing it to an existing ‘gold standard’ measure.” (Rebecca K. Meagher 2009).

Out of 103 recorded in-life observations, only five were not observed by ≥2 experts. This was on two occasions anxiety, on two occasions limb stiff and once twitches. All other 98 observations (81 primary and 17 optional) that were entered in-life were also seen on video by at least two experts. Based on these results, we conclude a high criterion validity of expert ratings on video compared to the ‘gold standard’ in-life observations.

For the construct validity of the observer ratings, we compared these to the expert observations (Fig 3 in the manuscript). Since the expert observations have a high criterion validity compared to in-life, they are considered as the ‘best approximate standardized test’ to which we compare the observer ratings.

---

## [Decision Letter · Decision Letter 2]

Application of video surveillance in preclinical safety studies in Canines: understanding the interobserver reliability and validity to recognize clinical behavior.

PONE-D-24-24225R2

Dear Dr. Kopljar,

We’re pleased to inform you that your manuscript has been judged scientifically suitable for publication and will be formally accepted for publication once it meets all outstanding technical requirements.

Kind regards,

Cord M. Brundage, D.V.M., Ph.D.

Academic Editor

PLOS ONE

Reviewers' comments:

Reviewer's Responses to Questions

**Comments to the Author**

1. If the authors have adequately addressed your comments raised in a previous round of review and you feel that this manuscript is now acceptable for publication, you may indicate that here to bypass the “Comments to the Author” section, enter your conflict of interest statement in the “Confidential to Editor” section, and submit your "Accept" recommendation.

Reviewer #2: All comments have been addressed

2. Is the manuscript technically sound, and do the data support the conclusions?

Reviewer #2: Yes

3. Has the statistical analysis been performed appropriately and rigorously? 

Reviewer #2: Yes

4. Have the authors made all data underlying the findings in their manuscript fully available?

Reviewer #2: Yes

5. Is the manuscript presented in an intelligible fashion and written in standard English?

Reviewer #2: Yes

6. Review Comments to the Author

Reviewer #2: The authors have addressed my primary concerns of the number of experts described for establishing the necessary and optional observations for COs as well as the availability of raw data in the manuscript.

The authors have addressed my editorial recommendations.

The authors have addressed Reviewer #3’s question of the types of validity that were assessed by the study aims.

7. PLOS authors have the option to publish the peer review history of their article (what does this mean? ). If published, this will include your full peer review and any attached files.

**Do you want your identity to be public for this peer review?** For information about this choice, including consent withdrawal, please see our Privacy Policy .

Reviewer #2: No

---

## [Editor Report · Acceptance letter]

PONE-D-24-24225R2

PLOS ONE

Dear Dr. Kopljar,

I'm pleased to inform you that your manuscript has been deemed suitable for publication in PLOS ONE. Congratulations! Your manuscript is now being handed over to our production team.

Kind regards,

on behalf of

Dr. Cord M. Brundage

Academic Editor

PLOS ONE